# Urinary Incontinence and Its Association with Physical and Psycho-Cognitive Factors: A Cross-Sectional Study in Older People Living in Nursing Homes

**DOI:** 10.3390/ijerph19031500

**Published:** 2022-01-28

**Authors:** Pau Farrés-Godayol, Javier Jerez-Roig, Eduard Minobes-Molina, Meltem Yildirim, Miriam Molas-Tuneu, Anna Escribà-Salvans, Sandra Rierola-Fochs, Montse Romero-Mas, Miriam Torres-Moreno, Laura Coll-Planas, Joanne Booth, Maria Giné-Garriga

**Affiliations:** 1Research Group on Methodology, Methods, Models and Outcome of Health and Social Sciences (M3O), Faculty of Health Sciences and Welfare, University of Vic-Central University of Catalonia (UVic-UCC), 08500 Vic, Spain; pau.farres@uvic.cat (P.F.-G.); eduard.minobes@uvic.cat (E.M.-M.); meltem.yildirim@uvic.cat (M.Y.); miriam.molas@uvic.cat (M.M.-T.); anna.escriba1@uvic.cat (A.E.-S.); sandra.rierola@uvic.cat (S.R.-F.); montse.romero@uvic.cat (M.R.-M.); miriam.torres@uvic.cat (M.T.-M.); laura.coll@uvic.cat (L.C.-P.); 2Research Centre for Health, School of Health and Life Sciences, Glasgow Caledonian University, Cowcaddens Road, Glasgow G4 0BA, UK; jo.booth@gcu.ac.uk; 3Department of Physical Activity and Sport Sciences, Faculty of Psychology, Education and Sport Sciences (FPCEE) Blanquerna, Ramon Llull University, 08022 Barcelona, Spain; mariagg@blanquerna.edu

**Keywords:** older people, nursing home, incontinence, urinary incontinence, pelvic health, sedentary behaviour, physical health, faecal incontinence, low urinary tract symptoms

## Abstract

Urinary incontinence (UI) is a common geriatric syndrome affecting bladder health and is especially prevalent in nursing homes (NHs). The aim of the study was to determine the prevalence of UI and its associated factors in five Spanish NHs. UI (measured with Minimum Data Set 3.0), sociodemographic, and health-related variables were collected. Chi-square (or Fisher’s) or Student’s *t*-test (or Mann Whitney U) for bivariate analysis were used, with Prevalence Ratio (PR) as an association measure. The prevalence of UI was 66.1% (CI:95%, 53.6–77.2) in incontinent (*n* = 45, mean age 84.04, SD = 7.7) and continent (*n* = 23, mean age 83.00, SD = 7.7) groups. UI was significantly associated with frailty (PR = 1.84; 95%CI 0.96–3.53), faecal incontinence (PR = 1.65; 95%CI 1.02–2.65), anxiety (PR = 1.64; 95%CI 1.01–2.66), physical performance (PR = 1.77; 95%CI 1.00–3.11), and cognitive state (PR = 1.95; 95%CI 1.05–3.60). Statistically significant differences were found between incontinent and continent NH residents for limitations in activities of daily living (ADL), mobility, quality of life, sedentary behaviour, and handgrip strength. It can be concluded that two out of three of the residents experienced UI, and significant associated factors were mainly physical (sedentary behaviour, frailty, physical performance, ADL limitations, mobility, faecal incontinence, and handgrip strength) followed by psycho-cognitive factors (cognition, anxiety, and quality of life).

## 1. Introduction

The total population of older adults in the European Union (EU), defined as those aged 65 years or older, is projected to increase significantly and will reach 129.8 million by 2050. During this transition, the number of people in the EU aged 75–84 years is projected to expand by 56.1%, while the number aged 65–74 years is expected to increase by 16.6% [1]. This complex process is caused by low fertility rates, increased life expectancy, and, in some cases, migratory patterns. This transition to an aged demographic pyramid in the coming decades will represent a challenge that will have to be managed in every country to provide health and social coverage to many older adults with multimorbidity, due to an increase of chronic conditions associated with age [2,3].

Lower urinary tract symptoms (LUTS) are a wide variety of symptoms that indicate poor bladder health. These symptoms can be divided into three groups: (a) storage symptoms, experienced during the storage phase of the bladder cycle and include daytime frequency, urinary incontinence (UI) and nocturia; (b) voiding symptoms experienced during the voiding phase including slow stream, hesitancy, and straining to start micturition; and (c) post-micturition symptoms experienced immediately after voiding, including incomplete emptying and post-micturition dribbling [4]. LUTS are highly prevalent world-wide but induce low levels of medical consultation [5,6,7,8]. UI is twice as prevalent in women than men, due largely to the impact of pregnancy, childbirth, and menopause, and their possible effects on pelvic organs and pelvic floor muscles [5,6,7,8,9]. LUTS and UI can happen at any age, but they are more common in older age. The prevalence is expected to increase with expected increases in mean life expectancy over the next decades, together with economic and psychosocial impacts on health care systems [2,8,9].

UI is the objectively demonstrable involuntary loss of urine that increases the subject’s frailty, their physical inactivity, their risk of falls, and their immobility; decreases their functional independence; and has negative physiological effects due to hygienic problems [4,9,10,11,12,13,14,15,16,17]. Also, UI greatly impacts on psychological health, severely affects normal social interaction and leisure activities, increases the risk of self-imposed isolation and cognitive impairment, and decreases satisfaction and quality of life (QoL) [10,18,19,20,21,22]. There are five types of UI: (a) stress UI is the involuntary leakage of urine that occurs with increases in intra-abdominal pressure (e.g., with exertion, effort, sneezing, or coughing); (b) urgency UI is the involuntary leakage of urine that may be preceded or accompanied by a sense of urinary urgency; (c) mixed UI is the involuntary leakage of urine caused by a combination of stress and urgency UI; (d) overflow UI is the involuntary leakage of urine from an overdistended bladder; and functional UI is the involuntary leakage of urine due to environmental, cognitive, or physical barriers to toileting [23,24]. In the nursing home (NH) population, the functional UI type stands out, caused by the inability to move to the bathroom independently, whether due to a physical or cognitive problems such as dementia [17].

In the NH population, UI is strongly associated with cognitive decline, inactivity, immobility, and impairment in activities of daily living performance that could lead to a decrease in physical activity, and an increase of sedentary behavior (SB) [20,21,25]. SB is considered a risk factor for cardiovascular disease, metabolic disease, obesity, frailty, disability, psychological disorders, and mortality decline [26,27,28,29,30,31,32,33]. Concerning the relationship between SB and pelvic health, a previous study found an association between urgency UI and the average duration of SB bouts, and another study found that low levels of physical activity were associated with greater nocturia and nocturnal enuresis; both studies were carried out on community-dwelling older women [34,35]. In addition, many authors suggest that low levels of physical activity and prolonged patterns of SB could be direct risk factors for UI in older adults [33,35,36,37,38].

To our knowledge there is no evidence on the association between SB and UI in NH residents. Consequently, new research is needed to analyse this relationship, with the aim of developing strategies to approach UI to improve resident’s health and QoL and reduce the UI burden on social and health services. The main objective of the study is to determine the prevalence of UI and its associated factors in a cohort of NH residents. Also, we aimed to verify the prevalence of the different types of LUTS and UI, as well as their impact on residents’ QoL.

## 2. Materials and Methods

### 2.1. Participants and Procedures

An observational cross-sectional study was carried out in five nursing homes (NHs) of Osona (a central Catalonia County, Spain). This sub study is part of the OsoNaH project [39], registered in Clinical Trials (NCT04297904). The STROBE (STrengthening the Reporting of OBservational studies in Epidemiology) guidelines for cross-sectional studies were followed [40]. The data was extracted from January 2020 and had to stop in March 2020, due to the COVID-19 outbreak. We included:All residents aged 65 years or olderResidents who lived permanently in the NHs.Residents or their legal guardians consenting to participate in the study.The exclusion criteria were:Subjects in a coma or palliative care (short-term prognosis).Residents with no cognitive capacity to answer questionnaires.Hospitalization.

The first contact with the NHs was done by email and phone call to explain the project and to solve any queries. Then, the information sheet and consent forms for the study were sent to them, if they were interested in participating. Every NH director who accepted the participation of their centre in the project signed a formal consent. After that, the list of residents was obtained, and the residents were selected according to the inclusion/exclusion criteria. Then, a simple randomization with the IBM SPSS Statistics software (IBM Corp. Released 2021. IBM SPSS Statistics for Windows, Version 28.0. IBM Corp.: Armonk, NY, USA) was done and the selected residents or their legal guardians were informed about the project, and those who agreed to participate signed the informed consent. At that time, NH staff were informed about the project, and those who agreed to participate, also signed the informed consent. The participants were informed that in the case of fatigue, they could interrupt or stop the assessment whenever he/she wished. Furthermore, they could withdraw from the study at any time without giving any reasons.

The research team was trained, received standardised operating procedures, and the inter-rater reliability was evaluated with the calculation of the Kappa index and the interclass correlation coefficient (ICC) of the data from 20 residents. The CCI scores were greater than 0.75 for all physical tests. The results for these 20 residents were not included in the final total study sample. After the reliability calibration, a pilot study was performed with a separate sample of 36 residents, whose data were included in the final sample.

### 2.2. Variables and Instruments

The dependent variable in the study was the presence of UI (yes/no) according to the proxy, by the Section H item 3a of Minimum Data Set (MDS) version 3.0 [41]. The presence of UI and other bladder and bowel conditions, urinary catheters, and incontinence control programs were also reported by the MDS. Additionally, the international test Consultation on Incontinence Questionnaire Urinary Incontinence—Short Form (ICIQ-SF) [42] and the International Prostate Symptoms Score (IPSS) [43,44] was applied. The ICIQ-SF assesses the quantity, frequency of urine losses, and the impact of the UI on the individual’s quality of life (QoL). The type of UI was determined according to the MDS and the ICIQ UI-SF. Information on LUTS and QoL associated with UI were collected using the IPSS. To evaluate the presence of nocturia, the residents and their proxy were also asked about the number of times the resident got up during the night to urinate. Residents with an ostomy and bladder catheterization were categorised as incontinent. Item 1 of the ICIQ-SF was taken as self-reported presence of UI; UI characteristics, LUTS, and QoL related to the UI of residents with capacity to answer the questionnaires aimed to compare their self-reported answers with the answers of the NH staff. The ICIQ-SF and IPSS answers were compared with the answers provided by the resident proxy through MDS.

Sedentary behaviour and waking time movement behaviour (WTMB) were assessed by the gold standard ActivPAL3 activity monitor (PAL Technologies Ltd., Glasgow, UK), worn on the anterior medial right thigh that captured the data continuously for 7 consecutive days [45,46,47]. The following variables were extracted: waking hours; standing duration in hours, percent of waking time standing, walking duration in hours, percent of waking time walking, absolute time upright in hours, percent of waking time upright, sit to stand transitions, sit to stand transitions per hour, absolute time sitting in hours, percent of waking time sitting, sitting bouts <30′, sitting bouts per hour <30′, total time sitting in <30′ bouts in hours, percent of waking time in bouts <30′, sitting bouts 30–60′, sitting bouts per hour 30–60′, total time sitting in 30–60′ bouts in hours, percent of waking time in bouts 30–60′, sitting bouts >60′, sitting bouts per hour >60′, total time sitting in >60′ bouts in hours, percent of waking time in bouts >60′, and average duration of SB bouts in minutes.

Activities of daily living limitations were measured using the modified Barthel Index by Shah et al. [48,49]. This scale is meant to be used in the assessment of patient performance or degree of assistance required in self-care, sphincter management, transfers, and locomotion. Shah et al. retained the original 10 items but proposed five-point rating scales for each item to improve sensitivity to detecting change. The scale consists of 10 items scored with several points that relate to ADL where the final score is calculated by summing the points awarded to each item. The categories were: independence (80 points), slight dependence (70–79 points), moderate dependence (31–69 points), and severe dependence (0–30 points). The continence items were excluded as already carried out by Jerez Roig et al. and Prado Villanueva et al. [33,50].

Physical performance was examined using the Short Performance Physical Battery (SPPB) [51]; the handgrip strength was measured by a Hand dynamometer (JAMAR Plus Digital: Warrenville, IL, USA), with the resident in a sitting position and their elbow at 90° of flexion; three repetitions were done in each hand, and the highest value was considered the valid one. The results were assessed and adjusted to sex and body mass index [52]. The resident’s participation in the NH exercise programs (mobility, respiratory gymnastics, multicomponent, or psychomotricity) conducted by the NH staff, was registered. To assesses frailty, the Clinical Frailty Scale (CFS) was chosen, because it is a practical, valid, and efficient tool that is considered a solid predictor of institutionalization and mortality. It is organized in an ordinal scale of nine points with clinical descriptions and pictograms where the NH staff has to consider the information about cognition, mobility, functionality, and comorbidities according to the history and the physical examination of the resident to choose in which grade of the scale the resident is [53]. The Rivermead Mobility Index assesses functional mobility in gait, balance, and transfers in 14 self-reported items and 1 direct observation item that progress in difficulty, with dichotomous answers coded as 0 for a no, indicating an inability to perform the activity or measure, and 1 for a yes. The summing of the points for all items indicates the final score, where higher scores indicate better mobility performance [54] For the sarcopenia risk, the SARC-F questionnaire was used. This is a rapid tool for screening sarcopenia risk, based on five components: strength, assistance in walking, rise from chair, climb stairs, and falls. The score range from 0 to 10, with 0–2 points for each component, and the final score of 4 or higher is predictive of sarcopenia [55,56]. The QoL using the self-reported questionnaire Spanish Index EuroQoL 5D-5L [57] and the daily consumption of fluids was assessed by a 24 h fluid consumption diary. The fluid diary had all the approximate volume of drinks in millimetres, the type of drink, and whether the drink was caffeinated or not, in a 24-h period. The diary was completed by the residents and corroborated by the proxy respondent.

Sociodemographic and health information including age, sex, level of education, marital status, smoking and drinking habits, body mass index (BMI), and comorbidities whether or not related to urinary incontinence, were obtained from the NH registers and checked with the NH professionals. Retrospective hospitalizations and fractures in the last 12 months; urinary tract infections in the last 30 days; and weight loss, ulcers, and delirium episodes in the last year were recorded. The total number of medications in daily use was registered according to the NH records, for medication related to chronic diseases, as well as the types of medications, according to the Anatomical Therapeutic Chemical classification system (ATCCS), a drug classification system that classifies the active ingredients of drugs according to the organ or system on which they act and their therapeutic, pharmacological, and chemical properties [58]. Also, we categorized the drugs that increase or decrease micturition and assessed the residents’ daily medication using the ATCCS, for likely relationship to their UI. Nutritional status was assessed using the Mini Nutritional Assessment (MNA) test, a validated screening tool to help identify elderly patients who are malnourished or at risk of malnutrition. According to the results, individuals can be divided into three groups using threshold values of <17 for ‘malnourished’, 17–23.5 for ‘at risk of malnutrition’, and ≥24 for ‘normal nutritional status’, with a maximum total score of 30 points [59]. 

Psychosocial factors were considered in all residents: the type of NH where they lived (whether private or subsidized); number of months they had been living at the NH number of monthly visits from friends/family, according to the NH staff; as well as the Yesavage Geriatric Depression Scale (GDS), considered a reliable and valid tool for self-rating of symptoms of depression in older adults 65 years of age or older. Of the 15 answers, 10 indicated the presence of depression when answered positively, while the rest (question numbers 1, 5, 7, 11, and 13) indicated depression when answered negatively. Scores of 0–4 are considered normal, depending on age, education, and complaints; 5–8 indicates mild depression; 9–11 indicates moderate depression; and 12–15 indicates severe depression [60]. Cognitive status was assessed using the Pfeiffer Scale, which evaluates functions such as orientation, memory, concentration, and arithmetic. The instrument classifies older adults (over 65 years) according to their preserved mental function: mild, moderate, or severe cognitive impairment, considering the educational level of the person being evaluated [61]. For anxiety, the Hospital Anxiety and Depression Scale—Anxiety subscale: HADS-A was used. A seven-item anxiety subscale focused mainly on symptoms of generalized anxiety disorder where each item scores on a 4-point Likert scale giving a maximum score of 21 for anxiety. According to the results, the scoring categorization is 0–7 for no anxiety, 8–10 for doubtful cases, and ≥11 for definite cases [10,62]. Social networks were assessed through the Lubben Social Network Scale-6 items (LSNS-6). This short version of six items scale is a self-reported measure of social engagement including family and friends. The LSNS-6 is correlated with mortality, all cause hospitalization, health behaviours, depressive symptoms, and overall physical health. The score ranges between 0 and 30, with a higher score indicating more social engagement [63]. For loneliness, the 6-item De Jong-Gierveld Loneliness Scale was used. This questionnaire is a reliable and valid measurement instrument that can be used as a unidimensional overall loneliness measure as well as provide information about the emotional and/or social loneliness situation of respondents. It is an individual’s subjective evaluation of his or her social participation or isolation. In this six-item scale, three answers consider emotional loneliness and three social loneliness. There are negatively (one to three) and positively (four to six) worded items. On the negatively worded items, the neutral and positive answers are scored as “1”, and on the positively worded items, the neutral and negative answers are scored as “1”. Therefore, this gives a possible range of scores from 0 to 6, which means closer to zero is least lonely and closer to 6 means most lonely [64].

### 2.3. Statistical Analysis

The sample size and power analysis were calculated according to the association between the dependent variable (UI yes/no) and the average duration of SB bouts in minutes, since this was the most significant SB variable in a previous study [33]. Since there are no previous studies analysing UI and SB in institutionalized older adults, we used our data for the sample size calculation. Considering a mean SB bout of 57.00 min (standard deviation-SD: 58.13) among the incontinent group and 19.94 min (SD: 12.64) among the continent group, with a significance level of 0.05 and power of 0.80, a minimum sample of 42 individuals was necessary.

Regarding statistical analysis, data obtained during the study were coded at the end of the collection, processed, and analysed by the members of the research group. The statistical analysis was carried out with the IBM SPSS Statistics software (IBM Corp. Released 2021. IBM SPSS Statistics for Windows, Version 28.0. IBM Corp.: Armonk, NY, USA). First, descriptive analysis was undertaken indicating absolute and relative frequencies for categorical variables and mean and standard deviation (SD) for quantitative variables. Subsequently, bivariate analysis was applied through the Chi square test (or Fisher’s test when necessary) and the linear Chi square test in cases of dichotomous or ordinal variables. The normality of data was evaluated with the Kolmogorov–Smirnov test to determine their distribution; for the parametric variables, the Student’s *t*-test was used and for the non-parametric variables the Mann–Whitney U test was applied. As an association measure, the Prevalence Ratio (PR) was used, with a confidence level of 95%.

## 3. Results

The final sample consisted of 68 residents with a mean age of 83.6 (SD = 7.6) years, mostly women (80.9%), and with an average duration living in the NHs of 29.1 (SD 29.0) months. Fifteen people (22.0%) lived in a private NH and 53 (78.0%) in a subsidized NH. From the 68 residents, 66 (97.0%) had descendants, with a mean number of 1.6 (SD 1.5) descendants, and of the 55 women, 47 (69.1%) had given birth. Figure 1 shows the flow chart of the excluded participants of the study.

Regarding the presence of diagnosed medical conditions, all 68 (100.0%) residents were diagnosed with at least one chronic condition, with a mean number of 5.1 (SD 2.3) diagnosed medical conditions. More specifically: 45 (66.2%) presented arterial hypertension, 30 (44.1%) cardiac disease, 29 (42.6%) dementia, 23 (33.8%) diabetes, 23 (33.8%) dyslipidaemia, 22 (32.4%) kidney failure, 20 (29.4%) diagnosed depression, 14 (20.6%) lung disease, 12 (17.6%) mental disease, 12 (17.6%) Parkinson’s disease, 12 (17.6%) stroke, 12 (17.6%) circulatory disease, 11 (16.2%) digestive disease, 11 (16.2%) hypothyroidism, 10 (14.7%) cancer, 10 (14.7%) diagnosed osteoporosis, 7 (10.3%) anaemia, 7 (10.3%) arthrosis, 6 (8.8%) vertigo, 5 (7.4%) diagnosed anxiety, 5 (7.4%) chronic pain, 5 (7.4%) visual deficit, 3 (4.4%) epilepsy, 3 (3.4%) hiatal hernia, 3 (4.4%) low blood pressure, 2 (2.9%) hyperparathyroidism, 1 (1.5%) hyperthyroidism, 1 (1.5%) ataxia, and 1 (1.5%) sleep disorders. Concerning pelvic health, three (4.4%) individuals had prostatic hyperplasia and two (2.9%) diagnosed vaginal prolapse.

It was found that 48 (70.5%) residents had cognitive decline, 62 (91.1%) presented functional dependency, 62 (91.1%) decreased functional capacity and frailty, 50 (73.5%) had risk of sarcopenia, 3 (48.5%) had risk of malnutrition, 58 (85.2%) residents presented depressive symptomatology, and 46 (67.6%) presented loneliness. All residents were using medication, with an average of 8.3 (SD 10.4) medications per day, 52 (76.5%) residents were taking five medications or more and 42 (61.8%) were at risk of anticholinergic adverse effects. It was identified that 30 (44.1%) residents had fallen at least one time in the last year, 17 (25.0%) had episodes of delirium, 9 (13.2%) had lost weight in the last 12 months, 8 (11.8%) had skin lesions, and 2 (2.9%) had leg ulcers. Also, in the last 12 months, five (7.4%) were hospitalized and four (5.9%) had a bone fracture. Table 1 shows socio-demographic and health-related information.

Concerning SB and the waking time movement behaviour (WTMB) of the residents, their average upright time was 2.2 (SD 1.8) hours, sitting time was 8.8 (SD 1.6) hours, and the average duration of SB bouts was 44.6 (SD 50.9) minutes. Also, 43 (63.2%) residents followed a physical exercise program led by NH staff at least once per week. Most of the residents who participated in the physical exercise programs had UI. Table 2 shows more information about SB and WTMB.

Regarding pelvic health, 2 (2.9%) residents had a permanent catheter, 4 (5.9%) residents had experienced urinary tract infection in the previous 30 days, 60 (88.2%) residents took medication that decreased their micturition, 54 (79.4%) residents took medications that produce an increase in micturition, and 21 (30.9%) took laxatives. The total average fluid consumption was 1864.1 (SD 804.7) millilitres (mL): 12.19 (SD 56.61) mL of cold drinks with caffeine, 228.13 (SD 316.83) mL of hot drinks with caffeine, 1396.48 (SD 633.73) mL of non-caffeine cold drinks, 121.70 (SD 206.93) mL of non-caffeine hot drinks, and 2.34 (SD 13.88) mL of alcohol. Nocturia (1+ waking at night to urinate) was reported in 21 (30.8%) residents by the proxy and 31 (45.5%) residents self-reported nocturia with the IPSS questionnaire.

According to the MDS, 44 residents presented with UI, i.e., prevalence of 66.1% (CI: 95%, 53.6–77.2). Among these, 33 (48.5%) had UI for more than a year, 22 (32.4%) had urinary leakage day and night, and 31 (45.6%) had large amounts of urinary leakage. From the 26 (38.2%) that were following a toileting program, only 8.8% totally improved their continence with the program. Besides, 12 residents had FI, i.e., prevalence of 17.6% (CI: 95%, 10.3–28.3). From these 12 residents, 11 (16.1%) presented double incontinence. Twenty-three residents (33.8%) followed a toileting program, 12 (17.6%) had constipation, and 5 (7.4%) diarrhoea. The average of faecal evacuations was 1.3 (SD 0.8) per day. In the total sample, the average number of diapers per resident was 2.2 (SD 2.0) per day, and within the incontinent group, 3.2 (SD 1.8). Table 3 shows UI and FI-related variables, according to the MDS.

According to the ICIQ-SF, 35 (51.4%) residents self-reported UI at least once a week. Among these, 17 (25.0%) before reaching a toilet, 19 (27.9%) when he/she coughs/sneezes, 23 (33.8%) whilst asleep, 13 (19.1%) during physical activity/exercise, 10 (14.7%) after urination while already dressed, 7 (10.2%) for no obvious reason, and 6 (8.8%) all the time. Regarding the frequency of urinary leakage, 11 (16.1%) subjects had leakages at least one time per week and 24 (35.2%) had daily losses. The urine leakage was low in 13 (19.1%) subjects, moderate in 11 (16.1%), and large in 12 (17.6%) residents. According to the MDS questionnaire, 7 (10.2%) incontinent residents reported no urine leakage or refused to answer. The impact of UI on QoL was low (score 0–3) in 22 (56.4%) subjects, moderate (4–6) in 8 (20.5%), and high (7–10) in 9 (23.0%) residents.

In the information gathered from the IPSS, bladder symptoms reported at least once in the last 30 days were: urgency: 37 (54.4%), increased frequency: 36 (52.9%), incomplete emptying: 20 (29.4%), intermittency: 18 (26.4%), weak stream: 16 (23.5%), and straining: 15 (22.0%). For the QoL associated with prostatic symptoms, 10 (14.7%) were delighted, 7 (10.3%) pleased, 16 (23.5%) mostly satisfied, 10 (14.7%) mixed, 12 (17.6%) not satisfied, 6 (8.8%) unhappy, and 4 (5.9%) terrible. Table 4 shows the classification of bladder symptoms and UI according to the ICIQ-SF and IPSS.

Table 5 shows the results of the bivariate analysis, and Table 6 and Table 7 show the group comparisons of the quantitative variables grouped by the dependent variable, and the presence of UI (yes/no) according to the proxy respondent, by Section H item 3a of the MDS questionnaire, with a *p* value equal to or under 0.200. The two groups were: continent (*n* = 23, mean age 83.00, SD = 7.7) and incontinent group (*n* = 45, mean age 84.04, SD = 7.7). The variables that were significantly associated with UI were anxiety, physical performance, cognitive status, frailty, and FI.

Regarding the comparison between the continent and the incontinent groups, the differences between groups were statistically significant in the variables of QoL, ADL limitations, mobility, handgrip strength, and all the SB and WTMB variables (see Table 6 and Table 7).

**Table 6 ijerph-19-01500-t006:** Association between urinary incontinence (UI) and parametric variables with the independent samples *t*-test among 68 residents living in five nursing homes (NHs) from Osona, Spain (2020).

UI							
	Yes		No				
	Mean	SD	Mean	SD	Mean Difference	*t*	*p*
Waking duration (h)	10.76	1.55	11.90	0.71	1.13	3.58	<0.001 *
Absolute time sitting in events <30 min	1.72	1.55	2.57	1.14	0.85	2.01	0.050
% time sitting in events <30 min	14.09	10.93	20.60	7.89	7.35	2.03	0.048 *
% time sitting in events between 30–60 min	14.09	10.93	20.60	7.89	6.50	2.42	0.019 *
Number of events between 30–60 min bouts	2.21	1.88	3.58	1.35	1.37	2.97	0.005 *
Number of events between 30–60 min bouts per hour	0.20	0.15	0.29	0.10	0.08	2.09	0.041 *
Absolute time spent in events >60 min	5.67	3.34	3.62	2.21	−2.04	−2.60	0.012 *
Right hand Handgrip	12.56	6.18	18.67	8.16	6.10	2.77	0.005 *
Left hand Handgrip	11.21	5.11	18.12	7.16	6.91	3.61	0.001 *
Dominant hand Handgrip	12.57	6.03	18.35	8.13	5.77	2.98	0.014 *

Own elaboration. Note: *n* = sample; % = frequency; SD = standard deviation; *p* = *p* value; h = hours; min = minutes; * Statistically significant.

**Table 7 ijerph-19-01500-t007:** Association between urinary incontinence (UI) and non-parametric variables through the Mann–Whitney U test among 68 residents living in five nursing homes (NHs) from Osona, Spain (2020).

UI						
	Yes		No			
	Mean Rank	Sum of Ranks	Mean Rank	Sum of Ranks	*U*	*p*
Spanish Index EuroQoL 5D-5L	29.55	1300.00	42.52	978.00	310.00	0.010 *
Barthel	28.21	1269.50	46.80	1076.50	234.50	<0.001 *
Rivermead Mobility Index	27.64	1244.00	47.91	1102.00	209.00	<0.001 *
Absolute time spent walking (h)	22.29	758.00	33.41	568.00	163.00	0.012 *
% of waking time walking	22.29	758.00	33.41	568.00	163.00	0.012 *
Absolute time spent standing (h)	22.26	757.00	33.47	569.00	162.00	0.011 *
% of waking time standing	22.00	748.00	34.00	578.00	153.00	0.007 *
Absolute time spent upright (h)	22.18	754.00	33.65	572.00	159.00	0.009 *
% of waking time upright	22.06	750.00	33.88	576.00	155.00	0.007 *
Sit to stand transitions	22.76	774.00	32.47	552.00	179.00	0.028 *
Sit to stand transitions per hour awake	3.09	785.00	31.82	514.00	190.00	0.048 *
% of waking time sitting	29.94	1018.00	18.12	308.00	155.00	0.007 *
Number of <30 min bouts	23.04	783.00	31.91	542.00	188.50	0.045 *
Number of <30 min bouts per hour	23.00	782.00	32.00	544.00	187.00	0.042 *
Absolute time sitting in events between 30–60 min	22.09	751.00	33.82	575.00	156.00	0.008 *
% of time sitting in events >60 min	29.00	986.00	20.00	340.00	187.00	0.042 *
Average duration of sedentary behaviour bouts in min	28.91	983.00	20.18	343.00	190.00	0.048 *

Own elaboration. Note: *n* = sample; % = frequency; U = Mann–Whitney U; *p* = *p* value; h = hours; min = minutes; * Statistically significant.

Further data results can be found in the Appendix A section (Table A1, Table A2 and Table A3) with variables with a *p* value over 0.200.

## 4. Discussion

This study aimed to verify the prevalence of UI and its associated factors in a sample of NH residents from Osona (Spain), as well as report information on other pelvic health issues such as FI and bladder symptoms. The findings indicate that the prevalence of UI was high, approximately 66%. Physical health issues (physical performance, frailty, FI, ADL limitations, mobility, SB and dominant hand handgrip strength, and psycho-cognitive issues (anxiety, cognitive state, and quality of life) were significantly associated with UI.

When comparing other studies in NH residents with capacity to answer questionnaires, the prevalence of UI is higher than the one found by Jerez-Roig et al. (2015) among Brazilian NH residents, with a prevalence of 42% [65], but slightly lower than the Jachan et al. (2019) study, with a prevalence of 70% in German NH residents [13]. In these studies, the MDS was used to assess the UI, except Jachan et al. who used the ICIQ-SF. Regarding other information on pelvic health given by the MDS, we found a prevalence of FI of 35% and 30% for double incontinence (faecal and urinary). An existing systematic review found a median prevalence for FI of 42.8% and 65% for double incontinence [66]. Our prevalence results are lower for double incontinence, however, regarding FI, the results are between the ranges they found.

Regarding LUTS, the most prevalent ones were storage symptoms, i.e., urgency, increased frequency, and nocturia, followed by voiding symptoms, intermittency, straining, and weak stream; post micturition symptoms [incomplete emptying], more common in men, were less frequent. In line with our results, a study conducted in community-dwelling older adults from Korea also found that storage symptoms were more prevalent than voiding symptoms and much more prevalent than post-micturition symptoms [8]. For nocturia, residents tended to self-report its presence more often with the IPSS than did the NH staff using the MDS. These findings might be partially explained due to NH staff not recording residents’ voiding patterns and residents not having a voiding diary routine. Since every resident’s room has its own private bathroom, particularly for the residents with preserved autonomy, the NH staff might not be aware of who is getting up at night to urinate. This is a possible hypothesis that could explain why the residents reported higher nocturia than the NH staff [67,68].

The prevalence of the different types of UI differs between the self-reported IPSS and the NH staff report. According to the self-reported ICIQ-SF, the most prevalent type of UI was nocturnal, followed by stress, urgency, dribbling post-micturition, and indeterminate and continuous UI. Nevertheless, NH staff reported that the most prevalent type of UI was urgency, followed by UI due to cognitive decline; functional, indeterminate UI; stress; and finally, effort UI. These differences between self-reported and NH staff answers could partially be explained by the difficulties of the NH staff in classifying the UI types due to a lack of knowledge on pelvic health disorders. It may also be due to the culture of secrecy and profound sense of shame felt by those that suffer from UI, which makes it very difficult to talk about and seek help from NH staff or health professionals, because residents may feel uncomfortable, embarrassed, or ashamed, as previous studies have reported [69,70,71,72,73]. This sense of shame and secrecy from people with UI is well reported in previous studies and profoundly affects their QoL in the domains of dignity, autonomy, and mood [19,71,74].

Incontinent residents had worse self-reported QoL than continent residents. UI can occur rapidly and in large volumes, which severely affects normal social participation among affected people [18]. It also increases the risk of isolation and decreases satisfaction with life. As a result, a decrease in QoL and functional independence of the resident can be observed, which may lead to greater frailty. There are different risk factors that could cause this decline in QoL, including sex, age, dementia and mobility, and the embarrassment of leaking urine and being wet can make residents feel a loss of personal dignity [74]. DuBeau et al. (2007) also showed that frail, functionally, and cognitively disabled residents with UI experienced a decrease in QoL [22].

Our study results also show that approximately three quarters of the sample who self-reported UI answered that UI had a low-moderate impact on their QoL, a lower proportion compared to previous studies. A review in Brazilian NH residents found that the impact of UI in QoL was higher than in our study, with approximately 57% reporting low-moderate impact [75]. Similarly, Jachan et al. (2019) found that in 51% of German NH residents, incontinence had a low-moderate impact on their QoL, according to ICIQ-SF [13].

Regarding LUTS and their impact on QoL, our results showed that most residents have good QoL. To our knowledge, there is no evidence of the impact of LUTS on QoL in NH residents. The evidence in community-dwelling older adults shows that prostatic symptoms and their severity increases with age, and also their higher frequency of poor QoL [76,77,78]. However, our results did not coincide with these previous statements, although they were in line with the results of the studies done by Adegun et al. (2016) and Ojewola et al. (2016) who found that even mild symptomatology could be associated with poor QoL, whereas some severe symptomatology could be associated with good QoL [77,78].

Regarding SB and WTMB, the incontinent group showed statistically significant differences, with lower periods of time spent standing, walking, and in an upright position; lower duration of time spent sitting in bouts of <30 min and bouts between 30 and 60 min; as well as higher time spent sitting in bouts of >60 min, than the continent group. Regarding SB, our results showed that all residents spent 80% of their waking time sitting or lying. Our results were slightly lower compared with Reid et al. (2013) in Australian NH residents, where they found an average of 85% of waking hours sitting or lying, or much lower than Chan et al. (2016) results in Canadian NH residents, where they found 95.1% of waking time spent sedentary. In all the studies, the gold standard ActivPAL monitor was used over 7 consecutive days, except Chan et al. who only used the device over 3 days [79,80]. Researchers still do not identify a threshold for how many consecutive minutes of sitting are needed before health risks are increased, but previous evidence suggests that sitting for as little as 20 consecutive minutes affects cardiometabolic health [81]. In line with the evidence of our study results, previous studies found statistical significance between UI and almost 20% longer duration of sedentary bouts in community-dwelling older women [34,39]. These findings could reinforce the hypothesis that the main risks of SB are the duration of the sitting bouts rather than the total time spent sitting. Finally, evidence suggests that SB is a risk factor for UI in older adults. Hence, future interventions to increase physical activity and break long SB bouts could be beneficial in preventing these symptoms [15,33,82].

Regarding factors associated with UI, our results show that incontinent residents had greater decline in their physical health and in their psycho-cognitive health than the continent individuals. On psycho-cognitive health, UI as a geriatric syndrome, has been strongly associated with cognitive decline and higher anxiety and depression levels in previous studies [18,22]. Regarding physical health, researchers over the years have found a strong association between UI and ADL limitations, frailty, physical performance, and overall physical condition loss [12,14,21,25,65,83,84]. However, there is no research done in NH populations to explain the results between low handgrip strength in incontinent residents. However, Bag Soytas et al. (2021) found a positive correlation between low handgrip strength and quantitatively measured weak pelvic floor muscle strength in adult women, a characteristic related to multiple causes in the pathophysiology of UI in women [85,86]. This could be one of multiple possible explanations, however, more research is needed to explain it in different populations and in both sexes. In addition, fluid intake, mobility, and diuretic treatment may also influence diuresis and, therefore, UI. In our study sample, neither medication nor total fluid intake were found to be significantly associated with UI. It could be interesting for further research to examine effects of these variables on UI in a longitudinal study [19].

The main limitation of this study was the small sample size, due to the COVID-19 outbreak that interrupted the data collection in NHs in March 2020 and reduced the number of participants initially estimated for the project. Consequently, a multivariate analysis of the significant variables could not be performed. In addition, since individuals without capacity to answer questionnaires were excluded from our study sample, the results cannot be generalized to residents with cognitive decline. Despite this, our sample size has sufficient power to analyze the association between UI and the average duration of sedentary bouts, since the differences between groups were high. Studies with a larger sample size, including confounders in multivariate analysis, are required to specifically analyze the role of SB in pelvic health. Also, the cross-sectional design of this study prevents establishing any cause–effect relationships between the variables; longitudinal designs are necessary to analyze these pathways. On the other hand, this study is the first that investigated the association between UI and SB in NH residents (considering both sexes), using the gold standard device ActivPAL to accurately measure SB. To our knowledge, this is also the first study describing LUTS and analyzing the association between UI and handgrip strength in institutionalized older adults. These results may be useful for the creation of new strategies to prevent UI and the SB effects on the NH population.

## 5. Conclusions

Our study found a high prevalence of UI, demonstrating that it affects approximately two out of three NHs residents in Osona. Incontinent residents had lower results in their physical health and in their psycho-cognitive health than the continent individuals. Incontinent residents had worse self-reported QoL than continent residents, but the specific impact of UI and LUTS on QoL was diverse. The results of this study found an association between UI and the average duration of sedentary bouts instead of the total time spent sitting. These findings could suggest that the main SB risk factor for UI relies on the duration of the individual SB bouts rather than the total time sitting accumulated during the day, but further investigation is needed to confirm this hypothesis. 

## Figures and Tables

**Figure 1 ijerph-19-01500-f001:**
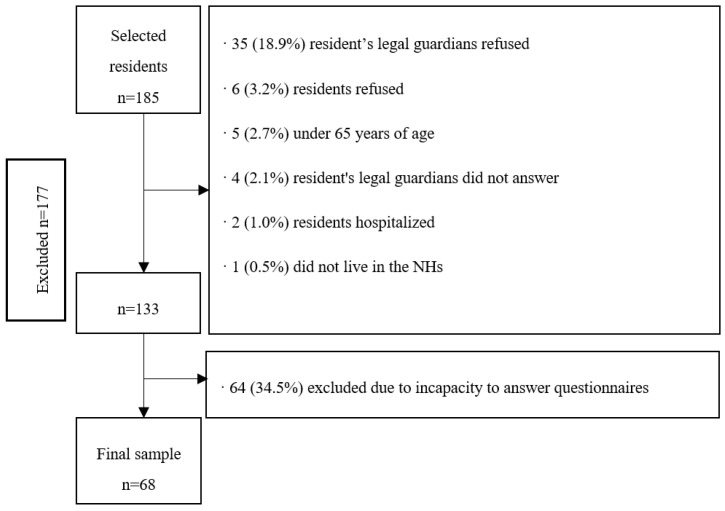
Flow chart of the sampling process of nursing home (NH) residents with capacity to answer questionnaires (Osona, Spain, 2020).

**Table 1 ijerph-19-01500-t001:** Socio-demographic and health-related information of five nursing homes (*n* = 68) from Osona, Spain (2020).

	*n*	Frequency (%)
Sex		
Male	13	19.1
Female	55	80.9
Education level		
Illiterate	21	30.9
Primary school	29	43.3
High school	5	7.4
College education	3	4.4
Unknown	10	14.7
Marital status		
Single	11	16.2
Married/dating	7	10.3
Divorced	2	2.9
Widow(er)	46	67.
Unknown	2	2.9
NH Type		
Private	15	22.0
Subsidized	53	78.0
Medication (ATC Classification) ^a^		
Group A	46	67.6
Group B	35	51.5
Group C	41	60.3
Group D	2	2.9
Group G	5	7.4
Group H	13	19.1
Group J	2	2.9
Group L	1	1.5
Group M	5	7.4
Group N	66	97.1
Group R	13	19.1
Group S	4	5.9
Group V	1	1.5
Body Mass Index (BMI)		
Under weight	11	16.2
Normal/Overweight	28	41.2
Obese	16	23.5
Unknown	13	19.1
Overall physical condition (Clinical Frailty Scale)		
Very fit	1	1.5
Well	13	19.1
Managing Well	4	5.9
Vulnerable	2	2.9
Mild Frail	16	23.5
Moderately Frail	25	36.8
Severely Frail	1	1.5
Very Severely Frail	0	0.0
Terminally ill	0	0.0
Physical performance (SPPB)		
Robustness	3	4.4
Prefrailty	9	13.2
Frailty	19	27.9
Disability	34	50.0
ADL limitations (Barthel)		
Independent	7	10.3
Slight dependency	10	14.7
Moderate dependency	35	51.5
Severe dependency	16	23.5
Cognitive state (Pfeiffer)		
Intact	20	29.4
Slight cognitive impairment	12	17.6
Moderate cognitive impairment	22	32.4
Severe cognitive impairment	14	20.6
Social Isolation (Lubben scale)		
No risk	51	75.0
Low risk	3	4.4
High risk	12	17.6
DNK/DNA	2	2.9
Anxiety (Hospital Anxiety and Depression Scale)		
Normal	52	76.5
Doubtful case	4	5.9
Definite case	11	16.2
DNK/DNA	1	1.5
Depressive symptoms (Geriatric Depression Scale)		
No	4	5.9
Positive	58	85.3
Suspected	6	8.8
Loneliness (The De Jong Gierveld Loneliness Scale)		
No	19	27.9
Positive	46	67.6
DNK/DNA	3	4.4
Nutritional state (Mini Nutritional Assessment)		
Normal nutritional status	28	41.2
At risk of malnutrition	32	47.1
Malnourshed	1	1.5
DNK/DNA	7	10.3

Own elaboration; Note: DNK/DNA = did not know/did not answer; ^a^ Drugs: N (Nervous System), A (Alimentary tract and metabolism), C (Cardiovascular system), B (Blood and blood forming organs), R (Respiratory System), H (Systemic hormonal preparations, excl. Sex hormones and insulins), G (Genito urinary System/sex hormones), M (Musculo-skeletal system), S (Ophthalmologicals), J (Antiinfectives), D (Dermatologicals), L (Antineoplastic agents), and V (Immunomodulating agents).

**Table 2 ijerph-19-01500-t002:** SB and WTMB information of five nursing homes (NHs) from Osona, Spain (2020).

	Mean	Standard Deviation (SD)
Waking hours	11.1	1.4
Standing duration (h)	1.8	1.5
% of waking time standing	15.9	13.2
Walking duration (h)	0.4	0.4
% of waking time walking	3.7	4.4
% of waking time upright	19.7	15.7
Sit to stand transitions	26.0	19.0
Sit to stand transitions per hour	2.0	2.0
% of waking time sitting	80.2	15.8
Sitting bouts <30′	22.0	19.0
Sitting bouts per hour <30′	2.0	2.0
Total time sitting in <30′ bouts (h)	2.0	1.4
% of waking time in bouts <30′	17.5	12.5
Sitting bouts 30–60′	3.0	2.0
Sitting bouts per hour 30–60′	0.2	0.1
Total time sitting in 30–60′ bouts (h)	1.8	1.2
% of waking time in bouts 30–60′	16.2	10.4
Sitting bouts >60′	2.0	0.9
Sitting bouts per hour >60′	0.2	0.1
Total time sitting in >60′ bouts (h)	4.9	3.1
% of waking time in bouts >60′	46.4	32.5

Own elaboration; Note: hours = (h); minutes = (min).

**Table 3 ijerph-19-01500-t003:** Urinary incontinence (UI) and fecal incontinence (FI) characteristics according to Mininum Data Set (MDS) among 68 residents living in five nursing homes (NHs) from Osona, Spain (2020).

	Sample (*n*)	Frequency (%)
Urinary toileting programme response		
No improvement	8	30.7
Decreased wetness	11	42.3
Completely dry (continent)	6	23.0
Unable to determine (or trial in progress)	1	3.8
UI		
Always continent	24	35.5
Occasionally incontinent	27	39.7
Frequently incontinent	10	14.7
Always incontinent	5	7.4
Not rated	2	2.9
Predominant type of UI		
Urgency UI	17	25.0
Stress UI	1	1.5
Cognitive decline UI	16	23.5
Functional UI	15	22.1
Indeterminate	2	2.9
FI		
Always continent	57	83.8
Occasionally incontinent	6	8.8
Frequently incontinent	3	4.4
Always incontinent	2	2.9

Own elaboration.

**Table 4 ijerph-19-01500-t004:** Self-reported information on urinary incontinence (UI) and bladder symptoms according to the International Prostate Symptoms Score (IPSS) and International Consultation on Incontinence Questionnaire-Short Form (ICIQ-SF) among 68 residents living in five nursing homes (NHs) from Osona, Spain (2020).

	Sample (*n*)	Frequency (%)
Final IPSS classification		
Mild (0–7)	36	52.9
Moderate (8–19)	21	30.9
Severe (20–35)	7	10.3
Unknown	4	5.9
Continence status by ICIQ-SF		
Continent (0)	26	38.2
Slight (1–5)	18	26.5
Moderate (6–12)	10	14.7
Severe (13–18)	8	11.8
Very severe (19–21)	3	4.4
Unknown	3	4.4

Own elaboration.

**Table 5 ijerph-19-01500-t005:** Association between UI (according to the MDS) and categorical independent variables with *p* value under 0.20 among 68 residents living in five nursing homes (NHs) from Osona, Spain (2020).

UI						
	Yes		No			
	Sample (*n*)	Frequency (%)	Sample (*n*)	Frequency (%)	*p*	PR (CI 95%)
Hypertension						
No	12	52.2	11	47.8	0.081	1.00
Yes	33	73.3	12	26.7		1.55 (0.82–2.95)
Dementia						
No	23	59.0	16	41.0	0.145	1.00
Yes	22	75.9	7	24.1		1.77 (1.03–3.02)
Kidney failure						
No	33	71.7	13	28.3	0.161	1.00
Yes	12	54.5	10	45.5		0.70 (0.37–1.32)
Delirium						
No	31	60.8	20	39.2	0.104	1.00
Yes	14	82.4	3	17.6		1.62 (1.01–2.60)
Group D drugs						
No	2	9.1	20	90.9	0.104 ^b^	1.00
1 or more	0	0.0	45	100.0		3.25 (2.25–4.68)
Anticholinergic risk						
No	14	56.0	11	44.0	0.133	1.00
Yes	31	73.8	11	26.2		1.33 (0.75–2.34)
Delirium						
No	31	60.8	20	39.2	0.104	1.00
Yes	14	82.4	3	17.6		1.62 (1.01–2.60)
Anxiety (HADS-A)						
No	29	59.2	20	40.8	0.014 ^b,^*	1.00
Yes	14	93.3	1	6.7		1.64 (1.01–2.66)
Exercise program participation						
Yes	31	72.1	12	27.9	0.176	1.00
No	14	56.0	11	44.0		0.70 (0.38–1.29)
SPPB						
Robustness—Prefrailty—Frail	16	51.6	15	48.4	0.018 *	1.00
Disability	27	79.4	7	20.6		1.77 (1.00–3.11)
Mini Nutritional Assessment (MNA)						
Normal	16	57.1	12	42.9	0.123	1.00
At risk—Malnourished	25	75.8	8	24.2		1.94 (1.00–3.74)
Pfeiffer Questionnaire						
Normal—Slight	17	53.1	15	46.9	0.032 *	1.00
Moderate—Severe	28	77.8	8	22.2		1.95 (1.05–3.60)
Clinical Frailty Scale (CFS)						
Very fit—Well—Managing Well—Vulnerable—Mildly Frail	20	55.6	16	44.4	0.003 *	1.00
Moderately Frail—Severely Frail—Very Severely Frail—Terminally ill	25	78.1	7	21.9		1.84 (0.96–3.53)
Faecal Incontinence						
No	34	60.7	22	39.3	0.006 *	1.00
Yes	11	91.7	1	8.3		1.65 (1.02–2.65)

Own elaboration. Note: * Statistically significant; ^b^ Fisher’s Exact Test.

## Data Availability

The data presented in this study are available on request from the corresponding author.

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
