# Peer review of "Urinary Incontinence and Its Association with Physical and Psycho-Cognitive Factors: A Cross-Sectional Study in Older People Living in Nursing Homes"

_ijerph, 2022, doi:10.3390/ijerph19031500_

Round 1

Reviewer 1 Report

Thank you very much for the opportunity to review. The work is interesting, but requires some refinement. Graphically, it is not transparent. Below are some suggestions to which please answer:

1. Please list the inclusion and exclusion criteria from the study in bullet points.

2. Please add tables or a graph that will clearly present data such as: age, gender, etc. Specify the average age of the respondents.

3. Do the authors have data such as comorbidities in the study group? This is important information for the reader and can affect urinary incontinence.

4. Please provide the results of the Barthel scale in the form of a graph or table.

5. What are the relationships between comorbidities and UI?

Good luck at a later stage of the publication.

Author Response

Thank you very much for the opportunity to review. The work is interesting but requires some refinement. Graphically, it is not transparent.

Response: The authors wish to thank you for your helpful and comprehensive feedback on our manuscript. The points raised are fair and compelling.

Below are some suggestions to which please answer:

  • Please list the inclusion and exclusion criteria from the study in bullet points.
  • Response: We thank the reviewer for this suggested addition and can confirm that the information has been added to the paper in the Participants and Procedures section.

  • Please add tables or a graph that will clearly present data such as: age, gender, etc. Specify the average age of the respondents.
  • Response: We thank the reviewer for this observation and have added the information accordingly in the first paragraph of Results and Table 1.

  • Do the authors have data such as comorbidities in the study group? This is important information for the reader and can affect urinary incontinence.
  • Response: We thank the reviewer for this observation. All data regarding comorbidities of the sample are fully described in the “results” section (second paragraph).

  • Please provide the results of the Barthel scale in the form of a graph or table.
  • Response: We thank the reviewer for this helpful suggestion. This information is included in Table 1.

  • What are the relationships between comorbidities and UI?
  • Response: We thank the reviewer for this observation and have added the information accordingly on the "appendix" section" the non-significant variables result (Table A1, A2 and A3).

Reviewer 2 Report

A major revision of the article should be carried out.
The changes to be made are as follows:
Abstract
-It should include that two groups are assessed, as well as the average age of the participants. 
Introduction
-The influence of gender on incontinence and the age at which incontinence begins to appear are not discussed.
Methodology
-The total number of participants is given.
-The sample is small.
-It is very risky to count water intake according to what the resident states. In general terms, I see that many aspects have been well quantified, such as the physical activity carried out, but there are many variables that depend on observation and are not quantifiable, and there may be a margin of error.
- What happens with patients who take diuretic drugs occasionally, or for a short period of time?
-How do you calculate those patients who, due to lack of staff at the time they wanted to go to the toilet, had no help?
-It should have been indicated how the groups were made.
-It would be advisable to make more than two groups for comparison, as age is a very important factor and should have been a variable to be categorised.
-Comorbidities related to urinary incontinence should have been noted. 
-Pelvic health is mentioned, but it is not indicated what types of exercises are done in the classes.
Results
There are a large number of variables analysed and the results of all of them are not apparent. There is a lot of descriptive data, and more correlations could have been made.
It would have been interesting to do an anova with several groups according to age and urinary incontinence. 
Discussion
There are few contrasts with studies that have studied similar aspects, even if not in institutionalised centres. 

Author Response

A major revision of the article should be carried out. Moderate English changes required:

Response: The article has been thoroughly reviewed by a native English researcher with extended experience (https://orcid.org/0000-0002-7870-6391).

The changes to be made are as follows:

Abstract:

  • It should include that two groups are assessed, as well as the average age of the participants. 
  • Response: We thank the reviewer for this suggested addition and can confirm that the information has been added to the “abstract”.

Introduction:

  • The influence of gender on incontinence and the age at which incontinence begins to appear are not discussed.
  • Response: We thank the reviewer for this suggested addition and can confirm that the information has been added to the “Introduction” section.

Methodology:

  • The total number of participants is given.
  • Response: We thank the reviewer for this observation, and confirm that we describe the participants in the "results" section in both text and flow chart (figure 1).

  • The sample is small.
  • Response: We thank the reviewer for this observation but according to the sample size calculation, we needed at least a sample size of 42 individuals, and we present a sample of 68 subjects. However, we recognize this issue as a limitation in the Discussion.

  • It is very risky to count water intake according to what the resident states. In general terms, I see that many aspects have been well quantified, such as the physical activity carried out, but there are many variables that depend on observation and are not quantifiable, and there may be a margin of error.
  • Response: We thank the reviewer for this observation. The fluid intake diary was completed by the residents and corroborated by the proxy; all this information was added to the "methodology" section.

  • What happens with patients who take diuretic drugs occasionally, or for a short period of time?
  • Response: We thank the reviewer for this observation. Regarding medication, the aim of the study was to gather information about the drugs used for chronic conditions, not about those occasionally prescribed, as frequently considered by most authors. Moreover, in our study the intake of diuretic drugs did not show statistical association with UI. These results are shown in the "results section" Table 5.

  • How do you calculate those patients who, due to lack of staff at the time they wanted to go to the toilet, had no help?
  • Response: We thank the reviewer for this observation, we considered this situation as functional UI, according to Abrams et al. (2010) as cited in the “Introduction” section, that describes it as the involuntary leakage of urine due to environmental, cognitive or physical barriers to toileting (1).

  • It should have been indicated how the groups were made.
  • Response: We thank the reviewer for this observation, this information has been extended in the "Methodology" section on the first paragraph of "2.2. variables and instruments" and in the "Results" section.

  • It would be advisable to make more than two groups for comparison, as age is a very important factor and should have been a variable to be categorised.
  • Response: We thank the reviewer for this observation, but because of the small sample size, if we had chosen more than 2 groups, we would have insufficient statistical power (2). We totally agree that age is a very important variable to consider, but its association with UI in our study was not significant (see appendix table A3).

  • Comorbidities related to urinary incontinence should have been noted. 
  • Response: We thank the reviewer for this observation, but the non-significant variables were not included in the main tables to make them clearer and more attractive for readers. The results showed no significant association with any comorbidity. For more information we have added the non-significant variables results (Table A1) to the "appendix" section" .

  • Pelvic health is mentioned, but it is not indicated what types of exercises are done in the classes.
  • Response: We thank the reviewer for this suggested addition and can confirm that more information has been added to the “methodology” section. We cannot be more specific with the exercise types (types, number, dosage, muscular groups and progression) because we did not collect this information since it was not the objective of our study.

Results:

  • There are a large number of variables analysed and the results of all of them are not apparent. There is a lot of descriptive data, and more correlations could have been made.
  • Response: We thank the reviewer for this observation, but the non-significant variables were not included in the results tables. For more information we have added the non-significant variables results (Table A1, A2 and A3) to the "appendix" section"

  • It would have been interesting to do an ANOVA with several groups according to age and urinary incontinence. 
  • Response: We thank the reviewer for this observation, but we only used the bivariate analysis because we had insufficient sample size for the multivariate analysis (2). Also, we chose the Chi square test and Fisher's test to assess the possible statistical association between UI and each independent variable.

Discussion:

  • There are few contrasts with studies that have studied similar aspects, even if not in institutionalised centres. 
  • Response: We thank the reviewer for this observation. There is scarce evidence on UI and its association with physical and psychosocial factors in NHs. However, we have added few sentences and new references in the “Discussion”.

References

  1. Abrams P, Blaivas JG, Stanton SL, Andersen JT. The standardisation of terminology of lower urinary tract function. The International Continence Society Committee on Standardisation of Terminology. Scand J Urol Nephrol Suppl [Internet]. 1988;114:5–19. Available from: http://www.ncbi.nlm.nih.gov/pubmed/3201169
  2. Suresh K, Chandrashekara S. Sample size estimation and power analysis for clinical research studies. J Hum Reprod Sci. 2012;5(1):7–13.

Round 2

Reviewer 2 Report

The article has improved significantly